# Effectiveness of Botulinum Toxin on Pain in Stroke Patients Suffering from Upper Limb Spastic Dystonia

**DOI:** 10.3390/toxins14010039

**Published:** 2022-01-05

**Authors:** Carlo Trompetto, Lucio Marinelli, Laura Mori, Luca Puce, Chiara Avanti, Elena Saretti, Giulia Biasotti, Roberta Amella, Filippo Cotellessa, Domenico A. Restivo, Antonio Currà

**Affiliations:** 1Department of Neurosciences, Rehabilitation, Ophthalmology, Genetics and Maternal and Children’ s Sciences (DINOGMI), Università degli Studi di Genova, 16132 Genova, Italy; ctrompetto@neurologia.unige.it (C.T.); laura.mori@unige.it (L.M.); luca1puce@gmail.com (L.P.); chiara.avanti@gmail.com (C.A.); sarettielena@yahoo.it (E.S.); giulia.biasotti@gmail.com (G.B.); robertaamella.RA@gmail.com (R.A.); filippo_cotellessa@hotmail.it (F.C.); 2IRCCS Ospedale Policlinico San Martino, Department of Neuroscience, Division of Neurorehabilitation, 16132 Genova, Italy; 3IRCCS Ospedale Policlinico San Martino, Department of Neuroscience, Division of Clinical Neurophysiology, 16132 Genova, Italy; 4Neurology Unit, Department of Medicine, Garibaldi Hospital, 95127 Catania, Italy; darestivo@libero.it; 5Academic Neurology Unit, Department of Medico-Surgical Sciences and Biotechnologies, Sapienza University of Rome, 04019 Terracina, Italy; antonio.curra@uniroma1.it

**Keywords:** upper motor neuron syndrome, spastic dystonia, spasticity, pain, botulinum toxin

## Abstract

This observational study aimed at investigating pain in stroke patients with upper limb spastic dystonia. Forty-one consecutive patients were enrolled. A 0–10 numeric rating scale was used to evaluate pain at rest and during muscle tone assessment. Patients were asked to indicate the most painful joint at passive mobilization (shoulder, elbow, wrist-fingers). The DN4 questionnaire was administered to disclose neuropathic pain. All patients were assessed just before and 1 month after incobotulinumtoxin-A treatment. Pain was present in 22 patients, worsened or triggered by passive muscle stretching. DN4 scored < 4 in 20 patients. The most painful joints were wrist–fingers in 12 patients, elbow in 5 patients and shoulder in the remaining 5 patients. Both elbow and wrist–fingers pain correlated with muscle tone. BoNT-A treatment reduced pain in all the joints, including the shoulder. We discussed that nociceptive pain is present in a vast proportion of patients with upper limb spastic dystonia. BoNT-A treatment reduced both spastic dystonia and pain in all the joints but the shoulder, where the effect on pain could be mediated by the reduction of pathological postures involving the other joints.

## 1. Introduction

In patients with upper motor neuron syndrome (UMNS), velocity-dependent hypertonia (greater resistance is experienced with fast stretches) is a common clinical sign [1]. The term “spasticity” is often used as a synonym of velocity-dependent hypertonia because neurophysiology defines spasticity as a “velocity-dependent increase in tonic stretch reflexes” [2]. We have recently shown that not only spasticity causes velocity-dependent hypertonia, but also another UMNS phenomenon called spastic dystonia [3,4,5]. Spasticity is a pathological stretch reflex, i.e., a stretch reflex evoked in fully relaxed muscles by the slow passive joint displacements used to assess muscle tone. In healthy subjects, stretch reflex cannot be evoked in these conditions. The basic trait of spasticity is that it is a dynamic phenomenon, which ceases when the passive movement stops [6]. On the contrary, spastic dystonia is not a pathological stretch reflex. It refers to the inability to voluntary silence muscle activity on command [7,8]. This inability leads to spontaneous tonic contractions that are stretch sensitive and produce pathological limb postures [9]. Overall, both spasticity and spastic dystonia manifest clinically with velocity-dependent hypertonia, but only spastic dystonia manifests with pathological resting postures [10].

Many studies have shown that pain is common after stroke, affecting up to 50% of stroke survivors, with the majority experiencing pain on a daily basis [11,12]. Several pain syndromes have been described, which can occur alone or in combination. Additionally, the pathophysiology of pain in stroke patients is complex, including both neuropathic and nociceptive mechanisms [13]. Muscle hypertonia is a risk factor for pain after stroke [14] and botulinum toxin-A (BoNT-A) proved an effective treatment for both hypertonia and pain [15]. Notwithstanding the number of studies, it has recently been highlighted that the existing literature on post-stroke pain is frequently challenging to interpret due to the methodological limitations of the studies and the in-homogeneity of participants [16]. This challenge fits particularly well with studies investigating pain in patients with increased muscle tone. In all these studies, spasticity was not differentiated from spastic dystonia since all hypertonic patients were considered to have “spasticity”. The actual impact of pain in patients with spastic dystonia is thus unknown. Studying pain in these patients may improve our understanding of the pathogenesis of pain in patients with velocity-dependent hypertonia and may help to deepen the knowledge on the effect of BoNT-A on pain in stroke survivors.

This study represents the first attempt to investigate pain in post-stroke patients affected by upper limb spastic dystonia. Our specific objectives were to assess pain prevalence, pain localization, pain type (nociceptive or neuropathic) and the effect of the toxin on pain.

## 2. Results

From 1 January 2021 to 31 July 2021, 41 post-stroke patients with upper limb spastic dystonia referred to our outpatient clinic. All 41 patients were enrolled in the present study (15 women; 65.1 ± 11.3 years; range 41–85 years). Demographics and clinical data are shown in Table 1. All 41 patients were treated with BoNT-A. All patients were clinically assessed just before BoNT-A treatment (time point 0, T0) and 1 month after (time point 1, T1).

### 2.1. T0 Clinical Assessment and BoNT-A Treatment 

According to the inclusion criteria, all patients showed pathological postures determined by muscles showing increased stretch reflex excitability (i.e., velocity-dependent hypertonia and/or increased deep tendon reflexes). In 34 patients (83%), both pathological postures of the shoulder, elbow, forearm, and wrist joints (Hefter’s patterns) and pathological postures of the fingers (affecting digits 2–5 and/or thumb) were found. Four patients (10%) presented only pathological finger postures (Hefter’s pattern 0), while the remaining 3 patients (7%), all with Hefter’s pattern IV, showed no pathological finger posture (F-and T-patterns 0) (Table 2).

Patients 1–9 scored > 0 on the pain numeric rating scale at rest (PNRS-rest); in two of them, pain was severe. Patients 1–22 scored > 0 on the pain numeric rating scale at passive mobilization (PNRS-mobilization); in 14 of them, pain was severe. In 20 patients, they scored < 4 in the DN4 questionnaire; in the remaining 2 patients, DN4 scored 4.

PNRS-mobilization scores were higher in women than in men (U = 95, male *n* = 26, women *n* = 15, *p* = 0.004). No difference in PNRS-mobilization scores was found between patients with ischemic and hemorrhagic stroke (U = 158, ischemic *n* = 26, hemorrhagic *n* = 15, *p* = 0.3), nor between patients with right- and left-affected sides of the body (U = 209, right *n* = 20, left *n* = 21, *p* = 1.0).

Age and PNRS-mobilization scores did not correlate (*n* = 41, Rho = 0.2, *p* = 0.3).

In 5 patients, shoulder was the most painful joint (shoulder-pain dominant). In four of them, the modified Ashworth scale (MAS) of shoulder adductors scored 0. Only the subject scoring MAS > 0 was treated with incobotulinumtoxin-A in shoulder adductors in addition to other muscles of the upper limb, while the remaining four patients were treated only in other muscles of the upper limb (see the Appendix A). In these 5 patients, PNRS-mobilization scores and MAS scores did not correlate (*n* = 24, Rho = 0.05, *p* = 0.8).

In five patients, the elbow was indicated as the most painful joint (elbow-pain dominant). In all of them, forearm flexors and/or forearm pronators MAS scored ≥ 2. In all of them, incobotulinumtoxin-A was injected in forearm flexors and/or forearm pronators. In these 5 patients, PNRS-mobilization scores and MAS score directly correlated (*n* = 24, Rho = 0.5, *p* = 0.02).

In 12 patients, the wrist and finger joints were indicated as the most painful ones (wrist–fingers-pain dominant). In all of them, the wrist and/or finger flexors MAS score was ≥ 2. In all of them, incobotulinumtoxin-A was injected in wrist flexors and/or finger flexors. In these 12 patients, PNRS-mobilization scores did not correlate with wrist–fingers flexors MAS scores (*n* = 31, Rho = 0.2, *p* = 0.3), but the 6 patients reporting severe pain scored higher MAS than those with mild pain, and the 19 patients with no pain (U = 36, high pain *n* = 6, low/no pain *n* = 25, *p* = 0.045).

### 2.2. T1 Clinical Assessment 

Considering all 22 subjects with PNRS-mobilization > 0 at T0, MAS scores at T1 decreased in forearm flexors (Z = −3.3, *p* = 0.0007), forearm pronators (Z = −2.8, *p* = 0.004), wrist flexors (Z = −3.4, *p* = 0.0005), and finger flexors (Z = −2.5, *p* = 0.01). Concomitantly, pathological postures changed in most patients. Both PNRS-rest (Z = −2.7, *p* = 0.008) and PNRS-mobilization (Z = −3.3, *p* = 0.0009) scores decreased at T1 (Table 3).

Considering the five shoulder-pain dominant patients, PNRS-mobilization scores decreased in four patients, although only one of them had received toxin treatment causing shoulder adductors MAS score to decrease.

Of the remaining 17 patients (5 elbow-pain dominant and 12 wrist–fingers-pain dominant), the MAS score remained unchanged in 5 patients, and the PNRS-mobilization score in 4 of them. In contrast, in the 12 patients with decreased MAS scores, also PNRS-mobilization scores decreased, except for 2 patients. However, no correlation was found between the decrease in MAS scores and the decrease in PNRS-mobilization scores (*n* = 17, Rho = 0.5, *p* = 0.08).

## 3. Discussion

This study aimed to investigate pain in stroke survivors manifesting upper limb spastic dystonia.

Among the 41 enrolled patients, 22 presented with upper limb pain (54%). Nine patients (22%) complained of pain at rest that increased during passive muscle stretching, with thirteen patients (32%) only during passive muscle stretching. The sensitivity to pain from passive muscle stretching points toward a nociceptive nature of pain, as confirmed by the DN4 questionnaire scores, that disclosed a neuropathic component of pain only in two patients. Therefore, pain accompanying upper limb spastic dystonia in stroke patients is predominantly nociceptive and has a very infrequent neuropathic component.

The intensity of pain proved greater in women than men. This finding is in line with much of the literature. Sex difference in pain perception has been well established, and women, on average, report more intense pain, more frequent pain, and pain of a longer duration than men [17].

Among the five elbow-pain dominant patients, four had Hefter’s IV pattern. Since this is a relatively rare pattern (only found in 10 out of the 41 enrolled patients), it appears to be more frequently associated with elbow pain than the other patterns involving elbow flexion. In our series, none of the patients exhibiting Hefter’s III pattern (23 of the 41 enrolled patients) were elbow-pain dominant. Since the main postural difference between Hefter’s III and IV patterns is the pronated forearm, reasonably the pain derives from the composite and concurrent mechanic action exerted at the elbow joint by forearm flexion and pronation. In these five patients, pain and tone scores correlated, indicating than both share some common element.

In the 12 wrist–fingers pain-dominant patients, muscle hypertonia was significantly more pronounced in those with severe pain than in those with mild or no pain. Nevertheless, the direct correlation between pain and muscle tone was not significant. Taken together, pain that results from elbow-pain dominant and wrist–fingers-pain dominant patients suggests that elbow pain and wrist–fingers pain in patients with upper limb spastic dystonia has some connection with the tone of the muscles acting on the painful joints. Whereas such connection is clear-cut for the elbow, it needs to be confirmed in a larger population for wrist and fingers. One possible factor for this difference is that the elbow is a multi-movement single-joint system, whereas wrist–fingers are a multi-movement multi-joint system, thus obeying distinct biomechanical constraints.

In patients with spastic dystonia, muscle hypertonia and pain likely exert a mutual influence at various levels. Tonic muscle contraction—the dominant trait of spastic dystonia that produces muscle hypertonia—may induce pain by itself. When the actively contracted muscle is passively stretched, pain increases. The same happens in healthy subjects who exert eccentric muscle contraction when they oppose to forceful stretching [18]. The abnormal postures sustained by spastic dystonia lead to secondary muscle changes, i.e., increased proportion of the connective tissue and muscle shortening. This transformation (muscle contracture) generates pain, especially during passive muscle stretching. Not only abnormal postures induced by spastic dystonia are a source of stress and damage for the muscle, but also for tendons and joints, that can act as an additional source of pain. Finally, skin changes induced by abnormal postures can be painful, as are palm ulcerations due to nail pressure in patients with closed fists [19]. On the other hand, pain can increase spastic dystonia, since all the positive UMNS phenomena are magnified by pain [10]. Since spastic dystonia is a more severe form of muscle hypertonia [4], pain may act by transforming spasticity into spastic dystonia, as we have often observed in clinical practice.

In this series of patients, incobotulinumtoxin-A treatment induced a significant pain reduction. Toxin analgesic effect in patients with muscle overactivity can be mediated by two mechanisms. Action at the presynaptic terminals of the spinal motor neurons blocks the release of acetylcholine, thus reducing excessive muscle contractions and contraction-related pain. Action at the nociceptive terminals of the peripheral afferent fibers blocks the release of neurotransmitters and neuropeptides involved in nociception, thus reducing nociceptive pain [20].

After incobotulinumtoxin-A treatment (Table 4), only three wrist–fingers-pain dominant and two elbow-pain dominant patients showed an unchanged MAS score; four of these patients also had unchanged pain scores. These negative observations corroborate the idea that in patients with spastic dystonia, the analgesic effect of botulinum toxin is mainly exerted by reducing muscle contraction.

Of the five shoulder-pain dominant patients, only one manifested hypertonia in muscles acting on the shoulder joint. In stroke patients, previous works suggest that the highly disabling condition known as “hemiplegic painful shoulder” has multifactorial pathogenesis. Shoulder pain has been associated with soft tissue injury related to subacromial bursitis, bicipital tendinopathy, glenohumeral subluxation, and rotator cuff pathology [21,22]. Four shoulder-pain dominant patients had no hypertonic muscles acting on the shoulder, yet in three of them, an incobotulinumtoxin-A injection in the muscles not acting on the shoulder reduced pain. Possible reasons for benefit are the following: the analgesic effect of the toxin is exerted remotely from the site of injection [23]; normalizing the posture of the elbow, wrist and fingers indirectly improves shoulder posture and ensuing pain; finally, a placebo effect.

## 4. Study Limitations and Strength

This is an observational study. The effect observed on pain could, at least in part, be due to a placebo effect. Spastic dystonia was assessed clinically to reproduce the most frequent real-world clinical setting. The sample size was relatively small, thus limiting the generalization of the results. Finally, we cannot exclude that at the time of the first assessment (T0), some patients were still under the effect of the previous injection, thus influencing pain intensity and prevalence.

Strength of this study is that the same investigator assessed all patients consecutively,

## 5. Materials and Methods

### 5.1. Patients’ Selection

This longitudinal observational study was conducted at neuro-rehabilitation outpatient clinic of the Department of Neuroscience, Ospedale Policlinico San Martino-IRCCS, Genova, Italy.

From 1 January 2021 to 31 July 2021, all stroke patients with upper limb spastic dystonia were consecutively enrolled. Spastic dystonia was diagnosed when the following two clinical signs were found: (1) upper limb pathological resting postures (e.g., elbow flexion); (2) velocity-dependent hypertonia and/or increased deep tendon reflexes in the muscle groups determining the observed postures (e.g., velocity-dependent hypertonia of the elbow flexors in a subject with flexion at the elbow).

The present study has been carried out in accordance with The Code of Ethics of the World Medical Association (Declaration of Helsinki) for experiments involving humans; written informed consent was obtained from all participants. This observational study was notified to the local ethics committee “Comitato Etico della Regione Liguria” and is scheduled for approval on 13 December 2021 (study ID 11983-n. 691).

### 5.2. Patients’ Assessment

Among the enrolled patients, those treated with BoNT-A were assessed twice, before (T0) and after treatment (T1). Those not treated were assessed only once (T0).

The tone of muscles acting at the shoulder, elbow, wrist, and finger joints was evaluated using the modified Ashworth scale (MAS), a 6-point scale ranging from 0 (no increase in tone) to 4 (limb rigid in flexion or extension) [24]. Muscle tone was assessed by the same examiner (CT) in all patients.

Upper limb postures were categorized by means of two different descriptors. Postures involving the shoulder, elbow and wrist joints were classified according to the 5 patterns described by Hefter and associates [25]. Finger postures excluding the thumb were classified as follows: finger-pattern (F-pattern) I: flexion of the first two phalanges with the hand closed in a fist (digits in contact with the palm); F-pattern II: flexion of the first two phalanges but no fist (digits not in contact with the palm); and F-pattern III: first phalanx in a neutral position (not flexed), and flexion of the second phalanx (third phalanx flexed or not) (clenched fingers). Finally, thumb postures were classified as follows: thumb-pattern (T-pattern) I: thumb flexion; T-pattern II: thumb opposition; and T-pattern III: thumb flexion and opposition [26].

By using the disability assessment scale (DAS) [27], four areas of disability were assessed (hygiene, dressing, limb position and pain) according to the following classification: 0 = no disability; 1 = mild disability not interfering significantly with normal activities; 2 = moderate disability (normal activities require increased effort and/or assistance); 3 = severe disability (normal activities limited).

Pain was evaluated using a numeric rating scale (PNRS), with 0 being “no pain” and 10 being “the worst pain imaginable” [28]. Patients were asked to rate the averaged pain perceived in the last week while the paretic limb was at rest (PNRS-rest). Then, they were asked to rate the pain they perceived during the muscle tone assessment maneuver just performed (PNRS-mobilization). Finally, patients were asked to indicate the most painful upper limb joint at passive mobilization (shoulder, elbow, or wrist–finger joints). A PNRS score of > 6 was considered to indicate severe pain.

The DN4 questionnaire was used to disclose neuropathic pain. The questionnaire includes seven symptoms and three physical examination items. A score of 1 is given to each positive item and a score of 0 to each negative item. Patients scoring ≥ 4 are considered to have neuropathic pain [29].

### 5.3. BoNT-A Treatment

Following our usual clinical practice, to relieve upper limb spastic dystonia each patient was considered for possible BoNT-A treatment. In those patients deemed candidates for treatment, incobotulinumtoxin-A (Merz Pharma GmbH & Co, Frankfurt am Main

Germany) was injected in the hypertonic muscles of the upper limb under ultrasound guidance. The muscles to be injected and toxin doses were determined in each single patient, according to the clinical picture, with the aim to reduce disability due to muscle hypertonia and/or pain, and to evaluate possible effects on pain itself.

### 5.4. Statistical Analysis

We used the Statistical Package for the Social Sciences (SPSS), SPSS Italia, Bologna, Italy, for all analyses. The Mann-Whitney test was used to assess the effect of gender (male/female), stroke type (ischemic/hemorrhagic) and the affected-body side (right or left) on PNRS scores. Furthermore, the Mann–Whitney test was used to compare MAS scores in patients with high pain (NRS > 6) in comparison to those with low pain or no pain (NRS < 7).

The Spearman rank test was used to correlate age and PNRS scores. The test was also used to correlate MAS and PNRS scores.

The Wilcoxon signed rank test was used to calculate changes in MAS and PNRS scores between T0 and T1.

## Figures and Tables

**Table 1 toxins-14-00039-t001:** Patients’ demographic and clinical features.

Patient	Gender	Age	Stroke Lesion Type	Affected Side	Time Since Stroke (Months)	Previous BoNT-A Injections	Last BoNT-A Injection (Months)
1	M	59	ischemic	R	45	7	9
2	F	47	ischemic	L	10	0	
3	M	84	ischemic	L	198	13	4
4	F	59	ischemic	R	100	10	4
5	M	65	ischemic	L	52	6	4
6	M	71	ischemic	L	216	21	3
7	F	63	haemorragic	L	29	3	10
8	F	52	ischemic	R	158	0	
9	M	73	haemorragic	R	68	11	3
10	M	84	ischemic	R	171	18	7
11	F	84	haemorragic	R	255	11	7
12	F	59	haemorragic	L	30	3	7
13	M	81	ischemic	L	18	0	
14	F	85	ischemic	R	61	5	7
15	F	51	haemorragic	R	111	0	
16	F	63	haemorragic	R	28	1	6
17	M	70	haemorragic	R	134	8	15
18	M	75	haemorragic	L	133	9	7
19	F	54	haemorragic	L	30	2	6
20	M	73	ischemic	L	99	15	5
21	M	64	ischemic	L	85	17	6
22	F	76	ischemic	L	39	4	4
23	M	71	ischemic	R	158	0	
24	F	72	haemorragic	R	60	11	3
25	M	41	ischemic	R	15	1	4
26	M	70	haemorragic	L	60	12	6
27	M	72	haemorragic	R	26	4	3
28	M	50	haemorragic	L	83	7	7
29	F	75	ischemic	L	46	7	6
30	M	65	ischemic	L	170	7	6
31	F	62	ischemic	L	52	8	5
32	M	64	ischemic	R	72	15	6
33	M	60	ischemic	L	65	12	5
34	M	48	ischemic	R	106	10	6
35	M	61	ischemic	R	76	18	3
36	M	58	ischemic	R	20	0	
37	M	57	haemorragic	L	74	7	4
38	M	56	ischemic	L	56	5	3
39	F	75	ischemic	R	246	0	
40	M	73	ischemic	R	87	14	3
41	M	48	haemorragic	L	71	3	7

M: male; F: female. R: right; L: left.

**Table 2 toxins-14-00039-t002:** Patients’ scores at T0.

	Shoulder-Pain Dominant	Elbow-Pain Dominant	Wrist-Hand-Pain Dominant	No Pain
	(*n* = 5)	(*n* = 5)	(*n* = 12)	(*n* = 19)
**MAS scores**				
ShA	0 (0–2)	1 (0−3)	0 (0−3)	0 (0−2)
FoF	2 (1−3)	3 (1−4)	2 (1−4)	1 (0−4)
FoP	2 (2−3)	3 (1−4)	0.5 (0−3)	2 (0−4)
WrF	3 (2−4)	4 (2−4)	3 (0−4)	2 (0−4)
FiF	3 (0−3)	3 (3−4)	4 (0−4)	3 (2−4)
**Hefter’ s pattern (n. patients)**				
No Pattern	1	0	1	2
Pattern I	0	0	1	2
Pattern II	0	0	0	0
Pattern III	3	0	7	13
Pattern IV	1	4	3	2
Pattern V	0	1	0	0
**Finger pattern (n. patients)**				
No Pattern	1	0	3	0
Pattern I	2	1	5	10
Pattern II	0	2	2	6
Pattern III	2	2	2	3
**Thumb pattern (n. patients)**				
No Pattern	0	1	7	4
Pattern I	2	1	4	7
Pattern II	1	3	1	3
Pattern III	2	0	0	5
**PNRS scores**				
PNRS-rest	2 (0−6)	2 (0−6)	0 (0−9)	0 (0−0)
PNRS-mobilization	7 (5−8)	8 (4−9)	6 (2−10)	0 (0−0)

MAS: modified Ashworth scale; scores are reported as median (range). PNRS: pain numeric rating scale; scores are reported as median (range). ShA: shoulder adductors; FoF: forearm flexors; FoP: forearm pronators; WrF: wrist flexors; FiF: finger flexors; PNRS: pain numeric rating scale.

**Table 3 toxins-14-00039-t003:** Patients’ scores at T1.

	Shoulder-Pain Dominant	Elbow-Pain Dominant	Wrist–Hand-Pain Dominant	No Pain
	(*n* = 5)	(*n* = 5)	(*n* = 12)	(*n* = 19)
**MAS scores**				
ShA	0 (0−1)	1 (0−3)	0 (0−2)	0 (0−2)
FoF	1 (1−1)	3 (0−4)	1 (0−3)	1 (0−3
FoP	2 (0−3)	2 (0−4)	0 (0−2)	0 (0−3)
WrF	0 (0−3)	2 (0−3)	2 (0−4)	0 (0−3)
FiF	0 (0−3)	3 (1−3)	3 (0−4)	2 (0−3)
**Hefter’ s pattern (n. patients)**				
No Pattern	3	0	5	6
Pattern I	0	0	1	2
Pattern II	1	0	0	0
Pattern III	0	1	3	9
Pattern IV	1	3	3	2
Pattern V	0	1	0	0
**Finger pattern (n. patients)**				
No Pattern	4	1	4	10
Pattern I	0	0	2	0
Pattern II	1	2	5	7
Pattern III	0	2	1	2
**Thumb pattern (n. patients)**				
No Pattern	3	1	12	14
Pattern I	0	1	0	0
Pattern II	1	3	0	5
Pattern III	1	0	0	0
**PNRS scores**				
PNRS-rest	0 (0−1)	0 (0−1)	0 (0−0)	0 (0−0)
PNRS-mobilization	5 (3−5)	8 (0−9)	3 (0−7)	0 (0−0)

MAS: modified Ashworth scale; scores are reported as median (range). PNRS: pain numeric rating scale; scores are reported as median (range). ShA: shoulder adductors; FoF: forearm flexors; FoP: forearm pronators; WrF: wrist flexors; FiF: finger flexors; PNRS: pain numeric rating scale.

**Table 4 toxins-14-00039-t004:** Toxin doses in injected muscle groups.

	Shoulder-Pain Dominant	Elbow-Pain Dominant	Wrist–Hand-Pain Dominant	No Pain
	(*n* = 5)	(*n* = 5)	(*n* = 12)	(*n* = 19)
**Toxin doses (UI)** **mean ± DS**				
ShA	100 (*n* = 1)	200 (*n* = 1)	100 (*n* = 1)	100 ± 0 (*n* = 2)
FoF	250 ± 0 (*n* = 2)	200 ± 41 (*n* = 4)	170 ± 71 (*n* = 10)	155 ± 44 (*n* = 14)
FoP	60 ± 22 (*n* = 5)	50 ± 0 (*n* = 5)	50 ± 0 (*n* = 6)	48 ± 7 (*n* = 14)
WrF	145 ± 62 (*n* = 5)	150 ± 41 (*n* = 4)	137 ± 62 (*n* = 10)	110 ± 35 (*n* = 13)
FiF	200 ± 121 (*n* = 5)	195 ± 99 (*n* = 5)	213 ± 72 (*n* = 10)	254 ± 107 (*n* = 19)

Toxin doses given as mean units ± SD, *n* = number of patients in which the muscle group has been injected. ShA: shoulder adductors; FoF: forearm flexors; FoP: forearm pronators; WrF: wrist flexors; FiF: finger flexors.

## Data Availability

The data presented in this study are available in the Appendix A.

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
