# Peer review of "Effectiveness of Botulinum Toxin on Pain in Stroke Patients Suffering from Upper Limb Spastic Dystonia"

_toxins, 2022, doi:10.3390/toxins14010039_

Round 1

Reviewer 1 Report

This paper reports on 41 patients with upper extremity spastic dystonia, 22 of whom had pain. The authors found that pain improved with botulinum toxin and that, except in those with predominant elbow pain, the presence or change in pain did not correlate with MAS score. 

There are already many publications on pain in upper extremity stroke,spasticity, and response of pain to botulinum toxin injection. A more complete review of that literature would be helpful along with expanded discussion on what new information is provided by this study.   The detailed information in the tables does not appear to be helpful; summary tables might be more useful.  It is also not clear why data was presented on 41 subjects, rather than just the 22 with pain.  Data should be provided, however, on toxin doses and muscles for the 22 injected patients.   On a minor note, table 3 is mislabeled; should probably be pain at T1 rather than T0.

Author Response

1) There are already many publications on pain in upper extremity stroke, spasticity, and response of pain to botulinum toxin injection. A more complete review of that literature would be helpful along with expanded discussion on what new information is provided by this study.

We have now provided a more complete review of the literature including four new references in the revised version of the manuscript (page 1, Introduction). The new introduction includes the following sentences:

Many studies have shown that pain is common after stroke, affecting up to 50% of stroke survivors, with the majority experiencing pain on a daily basis (Naess et al., 2012; Klit et al., 2011). Several pain syndromes have been described, which can occur alone or in combination. Also, the pathophysiology of pain in stroke patients is complex, including both neuropathic and nociceptive mechanisms (Harrison et al., 2005). Muscle hypertonia is a risk factor for pain after stroke (Wissel et al., 2010) and botulinum toxin-A (BoNT-A) proved an effective treatment for both hypertonia and pain (Wissel et al., 2016). Notwithstanding the number of studies, it has recently been highlighted that the existing literature on post-stroke pain is frequently challenging to interpret due to methodological limitations of the studies and in-homogeneity of participants (Plecash et al., 2019). This challenge fits particularly well with studies investigating pain in patients with increased muscle tone. In all these studies, spasticity was not differentiated from spastic dystonia since all hypertonic patients were considered to have “spasticity”.

New references

  • Naess H, Lunde L, Brogger J. The effects of fatigue, pain, and depression on quality of life in ischemic stroke patients: the Bergen Stroke Study. Vasc Health Risk Manag. 2012;8:407-13. doi: 10.2147/VHRM.S32780. Epub 2012 Jun 27. PMID: 22910531; PMCID: PMC3402053.

  • Klit H, Finnerup NB, Andersen G, Jensen TS. Central poststroke pain: a population-based study. Pain. 2011 Apr;152(4):818-824. doi: 10.1016/j.pain.2010.12.030. Epub 2011 Jan 26. PMID: 21272999.

  • Harrison RA, Field TS. Post stroke pain: identification, assessment, and therapy. Cerebrovasc Dis. 2015;39(3-4):190-201. doi: 10.1159/000375397. Epub 2015 Mar 5. PMID: 25766121.

  • Wissel J, Schelosky LD, Scott J, Christe W, Faiss JH, Mueller J. Early development of spasticity following stroke: a prospective, observational trial. J Neurol. 2010 Jul;257(7):1067-72. doi: 10.1007/s00415-010-5463-1. Epub 2010 Feb 6. PMID: 20140444; PMCID: PMC2892615.

2) The detailed information in the tables does not appear to be helpful; summary tables might be more useful.  

According to the suggestion of the reviewer we have now changes tables with new summary tables (Table 2 and 3)

3) It is also not clear why data was presented on 41 subjects, rather than just the 22 with pain. 

Since one aim of the study is to investigate the prevalence of pain in the patient cohort, all patients with spastic upper limb dystonia seen in our outpatient clinics from January 2021 to July 2021 were included in the study. Data prevalence are therefore given on the entire cohort (n=41).

In the revised version of the manuscript, we have now better specified this objective, to make it clear why we have selected all patients with upper limb spastic dystonia, without focusing only on those suffering from pain.

In the revised version of the manuscript, the last sentence of Introduction is:

This study represents the first attempt to investigate pain in post-stroke patients affected by upper limb spastic dystonia. Our specific objectives were to assess pain prevalence, pain localization, pain type (nociceptive or neuropathic) and the effect of the toxin on pain.

4) Data should be provided, however, on toxin doses and muscles for the 22 injected patients.

A new table (Table 4) was added, showing toxin doses and muscles for the 22 patients with pain.

5) On a minor note, table 3 is mislabeled; should probably be pain at T1 rather than T0.

We thank the reviewer for highlighting this mistake. We have now changed T0 to T1.

Reviewer 2 Report

Dear Authors, I have reviewed the following paper: Effectiveness of botulinum toxin on pain in stroke patients suffering from upper limb spastic dystonia. The findings are interesting, and the paper is well-illustrated. However, I have a few comments as follows:

Results section:

1) Please state, in the results section, that 41 patients were enrolled and that 22 of them were selected for BoNT treatment. 

2) In general, abbreviations are not properly used. Please note that if an abbreviation is defined in a document, it should be defined upon first usage and then used henceforth (with the Abstract – and often also the legends – being treated separately from the main text). Please correct the abbreviation for the following:

- Please define the abbreviation for T0 in your 2.1. T0 clinical assessment.

- In the second paragraph, please define the abbreviation for PNRS in PNRS-rest

- In the fourth paragraph, please define the abbreviation for MAS

- Please explain the abbreviation for T1 in the 2.2. T1 clinical assessment.

3) The fourth paragraph describes a BoNT injection being administered to a patient with MAS > 0 and shoulder pain. However, in four of the cases, the MAS scores were zero. Then, as a result, is not only one patient with shoulder pain treated with BoNT? This part is confusing.

4) 2.2. In Table 1, in the case of Last injection (months), what type of injection is it? If it is BoNT, it would be better to label it as the last BoNT injection.

And if it is a BoNT injection, is there any possibility that a patient who received BoNT 3 months ago will affect the results of pain reduction or decreased muscle tone after BoNT injection from this study (despite it being known that the effects of BoNT last for up to 6-8 weeks, then gradually decreases)?

5) Table 3 is titled Patients' scores at T0. I think it's probably T1. Please check carefully.

Discussion section:

1) In the 4th paragraph, the authors mention that the spasticity occurred unilaterally. Then how can the ultrasonographic and anatomic difference of the non-affected side be observed via ultrasonography?

2) In the 7th paragraph,  BoNT showed significant pain reduction. However, in the 9th paragraph, the author stated that pain reduction was not observed in patients with shoulder pain. Therefore, in the 7th paragraph, it seems correct to express that pain reduction was observed in all the patients who received BoNT, except for some cases among shoulder pain-dominant patients.

3) In the 9th paragraph, the authors state that BoNT had no effect on pain reduction in 3 out of 4 patients. The reason was simply expressed as such: the analgesic effect of the toxin is exerted remotely from the site of injection. But could the analgesic effect of toxins have been observed differently due to differences in pathophysiology or anatomy between the patients? In other words, it seems that a little more discussion on the difference in the analgesic effect of toxins between patients with shoulder pain is needed.

4) In the last sentence, there is a reference number 25, but there is no reference number 25 listed in the references list. Overall, the authors should review the reference number and citation position accurately once again.

Materials and Methods section:

1) As mentioned in the abstract,  please state that 22 patients were injected with BoNT.

Author Response

Results section:

1) Please state, in the results section, that 41 patients were enrolled and that 22 of them were selected for BoNT treatment. 

In the Material and Methods, we say that all the stroke patients with upper limb spastic dystonia seen from 1 January 2021 to 31 July 2021 were enrolled in the study (see Patients’ selection).

In the Results, we report that the patients enrolled were 41. All these 41 patients were treated with BoNT-A, because spastic dystonia was disabling in all of them. Therefore, all these 41 patients were clinically evaluated twice, once just before BoNT-A treatment (T0) and once after BoNT-A treatment (T1). At T0, 22 patients had PNRS-mobilization>0.

2) In general, abbreviations are not properly used. Please note that if an abbreviation is defined in a document, it should be defined upon first usage and then used henceforth (with the Abstract – and often also the legends – being treated separately from the main text). Please correct the abbreviation for the following:

- Please define the abbreviation for T0 in your 2.1. T0 clinical assessment.

In the revised version of the manuscript, abbreviation for T0 and T1 are defined in the fourth and fifth line of “Results”: “All patients were clinically assessed just before BoNT-A treatment (Time point 0, T0) and 1 month after (Time point 1, T1)”.

- In the second paragraph, please define the abbreviation for PNRS in PNRS-rest

In the revised version we have now defined PNRS in the results (Page 2, Lines 64-65).

- In the fourth paragraph, please define the abbreviation for MAS

In the revised version we have now defined MAS in the results (Page 2, Line 75).

- Please explain the abbreviation for T1 in the 2.2. T1 clinical assessment.

in the revised version we have now defined abbreviation for T0 and T1 in the fourth and fifth line of “Results”: “All patients were clinically assessed just before BoNT-A treatment (Time point 0, T0) and 1 month after (Time point 1, T1)”.

3) The fourth paragraph describes a BoNT injection being administered to a patient with MAS > 0 and shoulder pain. However, in four of the cases, the MAS scores were zero. Then, as a result, is not only one patient with shoulder pain treated with BoNT? This part is confusing.

Five patients reported that the main site of pain was the shoulder. All these five patients were treated with botulinum toxin (as were all the 41 patients investigated in this study). Among these 5 patients, only one had increased tone of shoulder muscles. In this patient, the toxin was injected into the shoulder adductors and also into other muscles of the upper limb. The other 4 patients had no hypertonia of shoulder muscles. Therefore, in them the toxin was NOT injected into the shoulder muscles, but only into the other muscles of the upper limb, according to their upper limb pathological postures (see supplementary materials, Table S1).

To avoid confusion, in the revised manuscript we have now modified paragraph fourth as follows:

In 5 patients, shoulder was the most painful joint (shoulder pain-dominant). In 4 of them, shoulder adductors MAS scored 0. Only the subject scoring MAS>0 was treated with incobotulinum toxin-A in shoulder adductors in addition to other muscles of the upper limb, while the remaining 4 patients were treated only in other muscles of the upper limb (see supplementary materials, Table S1). In these 5 patients, PNRS-mobilization scores and MAS scores did not correlate (n=24; Rho=0.05, p=0.8)”. (Page 2, lines 76-78)

4) 2.2. In Table 1, in the case of Last injection (months), what type of injection is it? If it is BoNT, it would be better to label it as the last BoNT injection.

In the revised version, in Table 1 we added “BoNT-A”: “Last BoNT-A injection (months)”

And if it is a BoNT injection, is there any possibility that a patient who received BoNT 3 months ago will affect the results of pain reduction or decreased muscle tone after BoNT injection from this study (despite it being known that the effects of BoNT last for up to 6-8 weeks, then gradually decreases)?

The referee raises a very good point. The effects of BoNT-A treatment on pain and hypertonia has distinct time course. Therefore, we cannot exclude that some patients might be under the effect of the previous injection, at least for pain, at the time of T0. However, we perform treatment when deemed appropriate (disabling hypertonia) and safe (respecting inter-injection time intervals).

We have now specified this important point in the revised version of the manuscript: "Study limitations and strength". In this chapter, we added the following sentence: “Finally, we cannot exclude that at the time of the first assessment (T0), some patients were still under the effect of the previous injection, thus influencing pain intensity and prevalence”. Page 8 lines 195-197

5) Table 3 is titled Patients' scores at T0. I think it's probably T1. Please check carefully.

We thank the reviewer for highlighting this mistake. We have now changed T0 to T1.

Discussion section:

1) In the 4th paragraph, the authors mention that the spasticity occurred unilaterally. Then how can the ultrasonographic and anatomic difference of the non-affected side be observed via ultrasonography?

Sorry, we did not evaluate the non-affected side, nor clinically neither ultrasonographically.

2) In the 7th paragraph, BoNT showed significant pain reduction. However, in the 9th paragraph, the author stated that pain reduction was not observed in patients with shoulder pain. Therefore, in the 7th paragraph, it seems correct to express that pain reduction was observed in all the patients who received BoNT, except for some cases among shoulder pain-dominant patients.

In paragraph 9 we did not say that “pain reduction was not observed in patient with shoulder pain” (please see next answer)

3) In the 9th paragraph, the authors state that BoNT had no effect on pain reduction in 3 out of 4 patients. The reason was simply expressed as such: the analgesic effect of the toxin is exerted remotely from the site of injection. But could the analgesic effect of toxins have been observed differently due to differences in pathophysiology or anatomy between the patients? In other words, it seems that a little more discussion on the difference in the analgesic effect of toxins between patients with shoulder pain is needed.

We are sorry for the lack of clarity. We try to explain better.

In paragraph 9, the results related to the 5 patients who indicated the shoulder as the main site of pain are discussed. As already mentioned, all these 5 patients were treated with botulinum toxin. Since only one of them had shoulder adductors hypertonia, only one patient was injected in shoulder muscles in addition to other upper limb muscles. After treatment, shoulder pain decreased in this patient. In the remaining 4 patients, no shoulder muscle was treated, because no shoulder muscle was hypertonic. These 4 patients were injected only in muscles acting on the elbow, wrist, and fingers. Nevertheless, after treatment, shoulder pain decreased in 3 out 4 of these patients.

4) In the last sentence, there is a reference number 25, but there is no reference number 25 listed in the references list. Overall, the authors should review the reference number and citation position accurately once again.

We apologize for the mistake. Inadvertently, reference 2 was numbered 1, thereby causing all subsequent references to be numbered -1. We thank the reviewer for noticing this source of serious confusion. We have now re-numbered the references.